# A multi-fidelity model benchmark for wake steering of a large turbine in a neutral ABL

Julia Steiner <sup>1</sup>, Emily Louise Hodgson <sup>2</sup>, Maarten Paul van der Laan <sup>1</sup>, Leonardo Alcayaga <sup>1</sup>, Mads Pedersen <sup>1</sup>, Søren Juhl Andersen <sup>2</sup>, Gunner Larsen <sup>1</sup>, and Pierre-Elouan Réthoré <sup>1</sup>

<sup>1</sup>Department of Wind and Energy Systems, Technical University of Denmark, Frederiksborgvej 399, 4000 Roskilde, Denmark <sup>2</sup>Department of Wind and Energy Systems, Technical University of Denmark, Anker Engelunds Vej 1, 2800 Kgs Lyngby, Denmark

Correspondence: Julia Steiner (julstei@dtu.dk)

Abstract. Wake steering is a promising control strategy for wind farm optimization, yet its effectiveness depends on the accuracy of underlying aerodynamic and structural models. In this study, we evaluate the predictive capabilities of models with varying fidelity for the IEA 22 MW reference turbine, considering both a single turbine and a two-turbine row with 5D spacing under conventionally neutral atmospheric boundary layer conditions. Results are benchmarked against large-eddy simulations (LES). All models reproduced qualitative trends in power and, where applicable, loads as a function of yaw angle and downstream position, but there was a large spread in quantitative agreement. The dynamic wake meandering (DWM) model implemented in Dynamiks gave very good predictions for mean power, acceptable results for blade and yaw bending Damage Equivalent Loads (DELs), but heavily underpredicted the tower bottom DELs compared to LES. RANS results from Ellip-Sys3D resolved asymmetric wake features, but with reduced magnitude, leading to increasing errors for power prediction with increasing wake deflection. Steady-state engineering models (PyWake and Fuga) performed reasonably well for power prediction in the aligned cases but showed increasing errors under yaw misalignment. None of the engineering models reproduced secondary steering. These findings highlight the limitations of the tested engineering and mid-fidelity models and emphasize the need for improved treatment of wake asymmetry, veer effects, and meandering physics to enhance reliability in practical optimization applications.

#### 15 1 Introduction

Wind turbines extract kinetic energy from the incoming wind flow, which creates wakes downstream of each turbine. These wakes are characterized by a velocity deficit and increased turbulence intensity (TI) (Porté-Agel et al., 2020). If a turbine is placed within the wake of another turbine, it produces less power than it would under undisturbed inflow conditions. In addition, the elevated turbulence within the wake leads to higher relative loading on the downstream turbine compared to the case of a simple reduction in wind speed without added turbulence. Furthermore, if only part of the downstream turbine is exposed to the wake, the structural loading can increase even further compared to a fully waked turbine (Herges et al., 2018).

Therefore, the relative placement of turbines is a key factor in wind farm design. More recently, different wind farm control strategies have been proposed that can be used during the farm's operation. These strategies aim to increase the total power

25

35

output of the farm and/or reduce the structural loading on individual turbines or the farm as a whole (Kheirabadi and Nagamune, 2019).

Static yaw control is one of the more promising control strategies that interests both industry and academia. The basic principle of static yaw control is the deflection of the upstream wake away from the downstream turbine using a predefined yaw misalignment angle to reduce the amount of waked inflow the second turbine experiences. This can potentially increase the power production of the downstream turbine, but it can also negatively affect the loads on both turbines. Hence, the yaw misalignment angle must be carefully chosen to ensure that the increase in power from the downstream turbine compensates for the loss in power of the upstream turbine, and that the changes in loading are acceptable (Meyers et al. (2022)). Indeed, how exactly the turbine loading is affected by flow control methods such as static wake steering, how large the yaw offset should be, and for which flow conditions does static flow control outperform other flow control methods, is still an active area of research. To answer those questions, aerodynamic models that can represent the effect of yawed inflow on both the rotor and the wake are necessary.

From an aerodynamic perspective, the flow behind a yawed wind turbine is rather complex. The misaligned inflow leads to a reduced thrust coefficient and a deflection of the wake, and the wake shape itself changes from circular to kidney-shaped for large yaw angles (Schottler et al., 2018). Close analysis of the cross-flow field in the wake of a yawed wind turbine with uniform inflow reveals two counter-rotating vortices at the top and bottom of the wake that lead to the deformation from a circular to a curled wake as visualized experimentally and also numerically by Mikkelsen (2004) and Howland et al. (2016). However, Vollmer et al. (2016) showed that atmospheric stability and the associated veer strongly influence the curling strength. They analyzed a single NREL 5 MW turbine under different stability conditions and yaw angles. For a yaw misalignment of  $|\gamma| = 30^{\circ}$  and at six rotor diameters downstream: (i) in a neutral boundary layer with around two degrees of veer over the rotor disk, the kidney shape is still visible and the magnitude of the vertical deflection relative to the tower position is similar for  $\pm 30^{\circ}$  yaw; (ii) in a stable boundary layer with about eight degrees of veer, the kidney shape is also visible, but the vertical deflection differs strongly between  $-30^{\circ}$  and  $+30^{\circ}$  yaw due to the strong veer; (iii) in a convective boundary layer, the wake appears largely unaffected by the yaw misalignment.

The asymmetry of the wake with a streamwise and lateral deficit also affects the downstream turbine. Fleming et al. (2018) demonstrated that the wake of a turbine aligned with the incoming wind and downstream of a yawed turbine also mildly deflects and coined the term secondary wake steering. A likely cause of this is that, on average, the lateral components of the deflected wake of the upstream turbine lead the downstream turbine to de facto also see a yawed inflow. Fleming et al. (2018) further noted that for a farm with multiple steered turbines, lateral deficits can also merge and further complicate the wake evolution.

At the early design stage of a wind farm, high-fidelity Computational Fluid Dynamics (CFD) simulations are generally not computationally feasible for comparing flow control strategies or optimizing yaw setpoints. This limitation is particularly restrictive for load predictions, where time-series data from Large Eddy Simulations (LES) would be required. Time-averaged Reynolds-Averaged Navier–Stokes (RANS) simulations are more tractable for power predictions, yet even these remain inefficient for early-stage design studies. Nevertheless, for specific farm layouts, surrogate or reduced-order models trained on

60

LES data have been successfully applied to wake steering and axial induction control (Hulsman et al., 2020; Debusscher et al., 2022).

For non-specific layouts, however, more computationally efficient approaches are necessary. To this end, a range of engineering wake models have been developed in the literature. These can be broadly divided into two categories: (i) models that rely on a symmetric circular wake description combined with a deflection model, sometimes with a projection into deflected coordinates (Jiménez et al., 2010; Bastankhah and Porté-Agel, 2016; Shapiro et al., 2018; Liew et al., 2023), and (ii) models that propose new deficit formulations explicitly accounting for wake curling (Martínez-Tossas et al., 2019; King et al., 2021; Branlard et al., 2022). In addition, Zong and Porté-Agel (2020) introduced an iterative momentum-conserving wake superposition method for streamwise and lateral deficits. Using the deflection model of Shapiro et al. (2018), they accurately reproduced both primary and secondary steering effects for wind-tunnel test turbines. As an alternative to traditional engineering models, linearized RANS approaches have also been applied to wake modeling and, more recently, adapted to also model wake deflection (Larsen et al., 2020).

To date, several publications have compared varying-fidelity models against high-fidelity CFD, LiDAR, and SCADA data. For example, Larsen et al. (2020) compared RANS, linearized RANS, DWM, and engineering model flow-field results with LiDAR measurements for a standalone V52 turbine. In the FarmConners benchmark, Göçmen et al. (2022) conducted a series of blind benchmark tests for wake steering, where lower-fidelity models were compared against both mean and time-series LES and SCADA data for single- and multi-turbine wakes with yaw offsets. These comparisons were quantified in terms of power predictions. The novelty of the present work lies in the level of detail of the comparison—covering flow, power, and loads—as well as in the breadth of models considered. Further, to date, in the literature, there are very few examples of a detailed load validation of the DWM against LES or measurement data for very large turbines. For load comparison against measurements, to the author's knowledge, the largest turbine used is a 6MW (Bernard et al., 2024a). For load comparison against LES, there are a few on the IEA 15MW turbines (Branlard et al. (2024); Rivera-Arreba et al. (2024)), but none on the IEA 22MW turbine.

This is important because larger turbines are more sensitive to the influence of shear and veer.

This paper aims to benchmark the whole toolchain of aerodynamic models developed at DTU against high-fidelity large eddy simulations (LES) for the large IEA 22MW reference turbine for primary as well as secondary wake steering at different angles, and also veered inflow for a conventionally neutral boundary layer (CNBL). The performance of the models is evaluated for power and, for the dynamic models, also loads. The array of tested models, ranging from lower to higher fidelity, is as follows: (i) the engineering models implemented in PyWake (Pedersen et al. (2023)) and Fuga (Ott et al. (2011)), (ii) a novel implementation of the Dynamic Wake Meandering (DWM) model in the open-source Dynamiks framework (Dynamiks repository), and finally (iii) the Reynolds-Averaged Navier-Stokes (RANS) solver implemented in EllipSys3D (van der Laan et al. (2024)).

Section 2 explains the setup and details the methodology used for all the models. Section 3 contains a detailed comparison between the models in terms of power and loads where applicable. Finally, the conclusions are given in Sect. 5.

## 2 Methodology

This section describes the methodology behind the setup in Sect. 2.1 and all the models in the remaining section.

#### 2.1 Setup

Two setups are considered. First, a single turbine with different prescribed yaw offset angles. Second, two turbines in a row with two different yaw setpoints for the first turbine and no yaw offset angle for the second turbine. The main parameters of the IEA22 MW turbine are specified in Table 1, and additional details are described by Zahle et al. (2024). The coordinate system is defined by the x-axis being aligned with the undisturbed flow direction at hub height, while the z-axis points upward. With this convention, the sign of the yaw angle is opposite to the right-handedness of the coordinate system about the z-axis.

| Turbine                |             | Setup 1        |                                                                  | Setup 2        |                           |
|------------------------|-------------|----------------|------------------------------------------------------------------|----------------|---------------------------|
| Type                   | IEA 22MW    | $x_{WT1}/D$    | [0,0]                                                            | $x_{WT1}/D$    | [0,0]                     |
| Diameter $D$           | 284 m       | $\gamma_{WT1}$ | $[-30^{\circ}, -20^{\circ}, -10^{\circ}, 0^{\circ}, 10^{\circ}]$ | $x_{WT2}/D$    | [5,0]                     |
| Hub height $h_{hub}$   | 170 m       |                |                                                                  | $\gamma_{WT1}$ | $[-30^{\circ},0^{\circ}]$ |
| Rotor tilt $\beta$     | $6^{\circ}$ |                |                                                                  | $\gamma_{WT2}$ | $[0^{\circ},0^{\circ}]$   |
| (a) Turbine properties |             |                | (b) Setup I                                                      | (c) Setup II   |                           |

**Table 1.** Turbine and setup properties.

# 2.2 LES

The LES results are obtained using EllipSys3D, a finite-volume incompressible Navier-Stokes solver initially developed by Michelsen (1992); Sørensen (1994), which solves the governing equations (including the potential temperature equation, as a conventionally neutral boundary layer (CNBL) is used as inflow) with a finite volume method in general curvilinear coordinates using a collocated grid. The pressure correction equation is solved with an extended SIMPLEC approach (Shen et al., 2003; Kobayashi and Pereira, 1991) using Rhie/Chow interpolation. Time advancement uses a second-order three-level implicit method with sub-iterations in each time step. Convective terms are discretised with a fourth-order central differencing scheme with a fourth-order dissipation term to reduce numerical instabilities (Wit and van Rhee, 2013). Sub-grid scale modelling is achieved using the AMD model (Abkar et al., 2016).

The inflow for all cases is based on a CNBL generated using an LES precursor simulation, conducted in a domain with size  $16640 \times 16640 \times 3000$  m, with grid resolution dx = dy = 4 m and dz = 20 m. The applied initial conditions are ground temperature  $T_{wall} = 288.15$  K; an initial temperature profile constant until 600 m height with perturbations up to 100 m height to encourage breakdown into turbulence, then a linear gradient of  $10^{-3}$  Km<sup>-1</sup> above; geostrophic wind values U = 9.48 ms<sup>-1</sup> and V = -0.59 ms<sup>-1</sup>; velocity profiles initialised with a log-law based profile with 600 m boundary layer height (Allaerts and Meyers, 2015); Coriolis parameter  $f_c = 1.1947 \times 10^{-4}$  s<sup>-1</sup>; and wall roughness  $z_0 = 0.001$  m. A wind direction controller

(Sescu and Meneveau (2014)) is active in order to achieve aligned flow at turbine hub height (170 m). Rayleigh damping is applied above a height of 2000 m (Klemp and Lilly, 1978). The precursor is run for 26 hours while the boundary layer develops, followed by a further period of 4500 s over which the three velocity components and temperature are extracted over a cross-stream plane in the domain centre, which are used as inflow to the successor simulation. The inflow profile is visualized in Fig. 1.

In the successor simulations, wind turbines are modelled using the actuator disc (AD) method (Mikkelsen, 2004), which is fully coupled to the aero-servo-elastic solver HAWC2 (Larsen and Hansen, 2025) (which has a Timoshenko beam element-based multi-body formulation and hence can account for non-linear effects) through the Dynamiks coupling framework (Dynamiks repository). The velocity components are extracted along the positions of the three rotating blades in EllipSys3D and passed to HAWC2, which calculates loads and deflections. These are transferred back to EllipSys3D and used to set the magnitude and position of the body forces that make up the AD. The AD body forces are applied in three overlapping 240° sections, where forcing decreases from a maximum at the blade location to zero at the neighbouring blades, which means that non-uniform loading can be captured. The two-way nature of the coupling means that the interaction between loading, deflections, and flow is also represented (Hodgson et al., 2021). HAWC2 also uses the DTU Wind Energy Controller, so the turbine varies rotational speed and pitch in response to its loading.

Successor simulations are conducted in a domain with size  $8520 \times 4970 \times 3000$  m  $(30D \times 17.5D \times 10.6D)$ , with a refined region stretching from the inlet to 20D downstream, laterally extending 3D in each direction from the domain centreline, and from the ground to 4D in height. Within the refined region, the grid resolution is dx = dy = dz = D/32 (Hodgson et al., 2023), and outside the cell size is stretched to the domain boundaries. The first turbine is placed 3.5D from the inlet, and for the two-turbine setup, a second turbine is placed 5D behind the first. Each simulation has 900 s which are used as spin-up, and the final 3600 s used for analysis.

# 135 **2.3 RANS**

The RANS results are computed by PyWakeEllipSys v5.4 (DTU Wind and Energy Systems, 2025), which uses a steady-state version of EllipSys3D. The k- $\varepsilon$ - $f_P$  turbulence model (van der Laan et al., 2015b) is employed coupled with an atmospheric boundary layer (ABL) inflow model including Coriolis forces and a simplified turbulent buoyancy source term that sets an ABL height implicitly (van der Laan et al., 2024). The latter is used to represent the stratification of the ABL, where lowering the ABL height results in a more stable ABL characterized by a larger wind shear and veer with respect to neutral conditions. The simplified turbulent buoyancy source term is only dependent on a prescribed Brunt-Väisälä frequency,  $\mathcal{N}$ , leading to an inflow model that does not require a temperature equation. The inflow is generated by a precursor simulation using EllipSys1D (van der Laan and Sørensen, 2017); the results are compared with LES in Fig. 1. The hub height wind speed and turbulence intensity based on k from LES (9.2 m/s and 3.8%) are obtained using a geostrophic wind speed G = 9.47 ms<sup>-1</sup>, and  $\mathcal{N} = 1.22 \times 10^{-3}$  s<sup>-1</sup>. The roughness length is set lower than LES,  $z_0 = 3 \times 10^{-5}$  m, to obtain a similar ABL height.

The turbine is modeled as rigid using an actuator disc (Réthoré et al., 2014) with an analytic force distribution model of Sørensen et al. (2020) that includes tangential forces and local effects of shear and veer. A precomputed lookup table

controls the AD aerodynamic coefficients as a function of the disc-averaged wind speed using single AD simulations without yaw and tilt (van der Laan et al., 2015a). The RANS simulations with the controller include a tilt angle of 6° and optional yaw angles.

The numerical domain of the single turbine simulations is a Cartesian box with dimensions  $(L_x, L_y, L_z) = (1025D, 1006D, 25D)$ . The turbine is placed near the horizontal center,  $(x,y,z) = (0,0,z_H)$ , and the flow around the turbine is resolved using a refined inner domain with dimensions: -5D 

185

195

and the wake evolves as the particles move downstream. Different settings are available for the background flow  $\underline{\tilde{U}}$ , but the settings that make the most sense are the local flow, including either all deficits or just the deficits upstream of the turbine from which the particle was emitted. Note that since the veer is included in the background flow, this also leads to a deflection of the particles due to the veer.

The advection velocity is further processed if the turbine is yawed or tilted. At the moment, two deflection models are available in Dynamiks, namely (i) the Hill Vortex model from Larsen et al. (2020), and (ii) the JiménezJiménez model from Jiménez et al. (2010). In this publication, we use the Hill Vortex model because it gave better results than the JiménezJiménez model, even when tuning the calibration parameter of the model for specific yaw angles. For the wake deflection with a yaw misalignment  $\gamma$  and a tilt angle  $\theta$ , the model writes as follows

$$\underline{\hat{U}}_{advection,j,k} = \underline{U}_{advection,j,k} - 0.4 \cdot \Delta U_{w,j,k} \begin{bmatrix} -\cos(\gamma) \cdot \cos(\theta) \\ \sin(\gamma) \cdot \cos(\theta) \\ \sin(\theta) \end{bmatrix}$$
(3a)

$$\Delta U_{w,j,k} = \frac{2}{R_w^2} \int_0^{R_w} \Delta U_j(\underline{x}_k) r dr \tag{3b}$$

where k indexes the particles belonging to turbine j with coordinates  $\underline{x}_{j,k}$ , and  $R_w$  is the wake expansion at the particle's location. For evaluating the advection velocity, the wake of the turbine from which the particle is emitted is excluded.

The most commonly used wake deficit model in the DWM context is the Ainslie model. It is derived from the steady-state RANS equations in cylindrical coordinates assuming: (i) symmetry with respect to rotation such that  $\frac{\partial}{\partial \theta} = 0$  and  $\tilde{U}_{\theta} = 0$ , (ii) that the shear layer between wake and freestream is thin such that the gradients of the mean flow fields are much bigger in the radial than in the axial direction, (iii) that the pressure gradient in axial direction is zero which only holds in the far wake, and (iv) that the Reynolds stresses can be modeled using a simple eddy viscosity model. The momentum equation in the axial direction and the mass conservation equation for a rotor-averaged inflow wind speed  $U_{\text{rotor}}$  will then look as follows

$$\tilde{U}\frac{\partial \tilde{U}}{\partial x} + \tilde{V}_r \frac{\partial \tilde{U}}{\partial r} = \frac{1}{r} \frac{\partial}{\partial r} \left( \nu_T r \frac{\partial \tilde{U}}{\partial r} \right),\tag{4a}$$

$$\frac{1}{r}\frac{\partial}{\partial r}\left(r\tilde{V}_r\right) + \frac{\partial\tilde{U}}{\partial x} = 0. \tag{4b}$$

with the boundary conditions

$$200 \quad \frac{\partial \tilde{U}(x,r)}{\partial r}\big|_{r=0} = 0, \tag{5a}$$

$$\lim_{r \to \infty} \tilde{U}(x,r) = U_{rotor},\tag{5b}$$

$$\tilde{U}(0,r) = \tilde{U}_0(r). \tag{5c}$$

The initial velocity deficit at the rotor  $\tilde{U}_0(r)$  can be calculated from a BEM model or a simple actuator disk model. Dynamiks has an interface to both HAWC2 and also the simple actuator disc turbines from PyWake. This publication uses the DWM model in combination with HAWC2 only. Since the pressure term in the momentum equation is dropped, the wake expansion and accompanying velocity deceleration in the near wake are not captured. To counteract this, the output of the actuator model is scaled before using it as an inlet boundary condition for the Ainslie model

$$\tilde{U}_0(r) = U_{\text{rotor}}(1 - (1 + f_U)a(\tilde{r})) \tag{6a}$$

$$r = \tilde{r}\sqrt{\frac{1-a}{1-(1+f_R)a}}\tag{6b}$$

where  $U_{\text{rotor}}$  is the rotor-averaged inflow wind speed,  $a(\tilde{r})$  is the radially varying induction obtained from the acutator model, and  $f_R = 1.1$  and  $f_U = 0.98$  are the tuning parameters for the wake expansion and velocity deceleration. Keck et al. (2012) claim that this makes the model accurate from 3D downstream.

The velocity deficit  $\Delta U_j$  of the turbine j that is used in the superposition is then given by  $\Delta U_j = U_{\text{amb},j} - \tilde{U}_j$ .

A simple mixing length model is used for the eddy viscosity as developed by Keck et al. (2012). It consists of two terms accounting for ambient and wake-induced turbulence, respectively

$$\nu_T = F_{amb}k_{amb}\text{TI}_{amb} + F_{wake}k_{wake}\max\left(\frac{1}{4}D_w(x)^2|\frac{\partial \tilde{U}}{\partial r}|, \frac{1}{2}D_w(x)(1 - \min_r(\tilde{U}(x,r)))\right)$$
(7)

where  $D_w(x)$  is the local wake diameter,  $TI_{\rm amb}$  is the turbulence intensity at the rotor, the parameters  $k_{\rm amb}=0.0914$  and  $k_{wake}=0.0216$ , and finally

$$F_{\text{amb}}(x) = \begin{cases} \frac{x}{4}, & \text{for } x 

$$\sigma^2 \approx 0.688\alpha \epsilon^{2/3} L^{2/3} \tag{9}$$

where  $\alpha$  is an empirical parameter and  $\epsilon$  is the dissipation of TKE.

The local velocity of the box is then multiplied by a scaling parameter  $k_{mt}$  as first proposed by Madsen et al. (2010)

$$k_{m_t}(r) = \left| 1 - \tilde{U}(r) \right| k_{m1} + \left| \frac{\partial \tilde{U}(r)}{\partial r} \right| k_{m2}$$
(10)

where the tuning parameters  $k_{m1}$  and  $k_{m2}$  are adapted from Branlard et al. (2024) as

$$f(\tilde{x}, f_{\min}, D_{\min}, D_{\max}, e) = f_{\min} + (1 - f_{\min}) \left(\frac{\tilde{x} - D_{\min}}{D_{\max} - D_{\min}}\right)^{e}$$
(11a)

$$k_{\text{def}}(x) = 0.6 f(\tilde{x}, 0, 0, 2, 1)$$
 (11b)

$$k_{\text{grad}}(x) = 3.0 f(\tilde{x}, 0, 0, 12, 0.65)$$
 (11c)

For the simulations presented in this paper, the LES precursor simulations are used to generate the background turbulence and mean flow for the most direct comparison. The precursor inflow planes are stacked together, and the mean velocity at hub height is the advection speed of the box. The wake-added turbulence box is computed as described above using one seed. As in the LES simulations, the same HAWC2 setup file and controller are used for the structural turbine model.

#### 2.5 PyWake




Steady-state flow conditions for various prescribed yaw angles were simulated using PyWake (Pedersen et al., 2023). The inflow conditions were selected to match the hub-height wind speed and turbulence intensity obtained from LES results. The simulation does not include blockage effects and propagates downstream a wake deficit modeled using the Supergaussian model with the recalibration according to Blondel (2023). Wake-generated turbulence is added on top of the prescribed background turbulence intensity using the model from Crespo and Hernández (1996). A root-of-summed-squares superposition method is used to combine multiple wakes. The wake deflection is modeled using the JiménezJiménez model with  $\beta = 0.1$  (Jiménez et al., 2010), an engineering wake deflection model empirically derived from LES data, which accounts for lateral displacement of wakes due to turbine yaw.

#### 250 **2.6** Fuga

The turbine wake evolution is modeled using a fast linearized CFD solver based on the Reynolds-Averaged Navier-Stokes (RANS) equations with an eddy viscosity closure. Fuga (Ott et al., 2011; Ott and Nielsen, 2014) assumes that the forcing from the turbine is small relative to the background shear flow and applies a first-order perturbation expansion. The governing equations are solved efficiently using a mixed spectral formulation, where a Fourier transform in the horizontal directions reduces




the system to a set of decoupled ordinary differential equations in the vertical coordinate. Lookup tables are precomputed and reused to account for different turbine positions and forcing without solving the full equations for each new case.

Non-linear effects of yaw-induced wake deflection are captured by adapting a linearization technique (Ott et al., 2019; Larsen et al., 2020). This involves a curvilinear coordinate transformation in which the transverse direction is shifted by a small displacement  $\lambda(x,y,z)$ , such that the new transverse coordinate  $\acute{y}=y-\lambda$  remains constant along streamlines. This transformation effectively removes the wake deflection in the new coordinates, only allowing it to be recovered upon returning to the original space. The displacement  $\lambda$  is computed from the lateral velocity field and the background streamwise velocity and stored as an additional lookup table.

The inflow conditions were designed to match the mean wind profile obtained from LES simulations, using a target wind speed at hub height of  $U_{hub}=9.2$  m/s and a prescribed surface roughness length of  $z_0=0.001$  m. These conditions are used to define the background velocity profile  $U^0(z)$  and the eddy viscosity K(z) via Monin-Obukhov similarity theory, with the stability function for momentum  $\phi_m(z/L)$  set for neutral conditions.

Since Fuga is linear by definition, wake superposition is also linear. This means the results of superimposed deflected wakes might be slightly affected by breaking the assumption of null deflection at the wind turbine rotor. In our case, the downstream turbine is not yawed, so the effect of this simplification on the resulting wind fields is negligible.

# 270 3 Results

The sections contain a comparison of the inflow profiles in Sect. 3.1, the results for the single turbine setup in Sect. 3.2, and finally the results for the two-turbine setup in Sect. 3.3.

#### 3.1 Inflow

Care is taken to closely match the inflow profiles for the different models' mean velocity, inflow direction, and turbulence intensity. Figure 1 shows the results at the inlet of the domain.

The wind direction controller in the LES precursor setup did not fully converge, so there is a slight offset in the wind direction at hub height of  $\Delta\phi=-0.5^{\circ}$ . Averaged over the rotor disk, this becomes  $\overline{\Delta\phi}=0.4^{\circ}$ . This offset in wind direction was applied in the remaining model simulations to ensure consistency between the setups. Half a degree of wind direction offset corresponds to roughly three degrees of yaw steering; hence, it was important to include it. Since Fuga and PyWake do not model veer, the wind direction remains constant with height.

For the comparison of the turbulence intensity (TI) to LES, total and streamwise formulations are used for RANS and the DWM, respectively:

$$TI = \frac{\sqrt{2/3k}}{U_{ref}} = \frac{\sqrt{1/3(\sigma_U^2 + \sigma_V^2 + \sigma_W^2)}}{U_{hub}}$$
(12a)

$$TI_{u} = \frac{\sigma_{U}}{U_{hub}} \tag{12b}$$



For the RANS model, the inflow turbulence is isotropic, which does not correspond to reality. This means the RANS model underpredicts the streamwise TI, but the total TI is close to the LES profile, since the TKE is fitted to match the LES. Hence, the total TI is used for comparison with LES for the RANS model. For the DWM model, only the streamwise part of the TI is fully approximated; hence, the streamwise TI is used for comparison to LES.

# Inflow Wind at hub height $U_{hub}$ 9.2 ms<sup>-1</sup> Turbulence intensity $TI_{hub}$ 4% Wall roughness $z_0$ 0.001 m Coriolis parameter $f_c$ 1.1947 × 10<sup>-4</sup> s<sup>-1</sup>, Boundary layer height $\delta_{BL}$ 900 m

**Figure 1.** Comparison of inflow profiles from different models matched to the LES precursor.

Table 2. Inflow properties

#### 3.2 Setup I: A single wind turbine

First, a visual comparison of the model predictions through horizontal slices of the streamwise velocity at hub height for the fully aligned case and the misaligned case with  $\gamma=-30^\circ$  is shown in Fig. 2. For both the aligned and the misaligned cases, the prediction of the far wake deficit and wake recovery rate seems reasonably well captured by all the models. For the near wake, there is more variation. The RANS model is close to the LES. The DWM model overpredicts the deficit close to the rotor per the boundary conditions for the deficit model, which neglect the gradual pressure recovery after the rotor, but is close to the LES mean from about 2D onwards. The linearized RANS solution from Fuga underpredicts the deficit in the near wake but starts to agree well with LES from about 5D onwards. At first glance, the velocity field at the rotor location in Fuga suggests that the turbine yaws in the wrong direction. However, this is simply an artifact of how the turbine forcing is implemented in Fuga: it uses a non-yawed disk that applies forcing in both the longitudinal and transversal directions. When looking further

upstream from the turbine, the blockage pattern shows that the rotation occurs in the correct direction. Finally, the engineering model implemented in PyWake predicts the flow deceleration behind the rotor due to wake expansion accurately in terms of magnitude, but the transition from near to far wake comes too late, and the deficit starts to agree with the LES from about 6D onwards. Since the mean wind direction misalignment over the rotor disk points downward and the wake moves upwards due to tilt in an area where the mean veer also points downwards, the wake should deflect a bit even in the aligned case. Fuga and PyWake do not model the veer, so they can only model the part of that effect related to the mean wind direction misalignment.

Figure 2. Time-averaged streamwise velocity at hub height for all models for (left) the aligned and (right) the yawed case with  $\gamma=-30^\circ$ .

Moving on to the TI, the RANS, PyWake, and DWM fields are compared to the LES fields; the results from Fuga are not shown since it does not predict TI. As explained in Sect. 3.1, differing TI formulations are used for the DWM and RANS/Py-Wake. The top plots of Fig. 3 depict the total turbulence intensity TI for the mean LES, RANS, and PyWake predictions. As expected, RANS overpredicts TI as compared to LES, but surprisingly, the overprediction is less pronounced for the yawed case, even though the deficit magnitude predictions are similar between the two. The WAT model in PyWake heavily overpre-

dicts the TI as compared to LES and RANS. The bottom plots of Fig. 3 show the streamwise turbulence intensity  $TI_u$  for the mean LES and DWM predictions. With the updated WAT model, at hub height, the DWM predictions are very close to the ones from LES, both in the near and far wake, even for the deflected case. The high TI region due to the tip and root vortices in the near wake persists for a bit longer in the LES than in the DWM. However, since the WAT formulation does not consider shear, below and above hub height, the TI is under- and overpredicted, respectively.

Figure 3. Turbulence intensity (TI) at hub height across five columns: (I)-(III) total TI from LES, RANS and PyWake, and (IV-V) streamwise TI from LES and the DWM model. On the (left) is the aligned case, and on the (right) is the yawed case with  $\gamma = -30^{\circ}$ .

To gauge more accurately how well the wake deflection is captured for different yaw misalignment angles, a wake tracking algorithm was run on the (time-averaged) flow fields for all models. The Constant Area wake center tracking algorithm from the SAMWICH Toolbox was run on the streamwise velocity fields. Other simpler wake-tracking algorithms were tested, but


they performed unreliably due to the asymmetry of the wake due to veer and curling. The waketracking algorithm identifies 320 velocity deficit isocontours and looks for the contour line that roughly corresponds to the rotor area. Then a weighted average over said area is used to calculate the wake center  $(y_{WC}, z_{WC})$  as

$$y_{\text{WC}} = \frac{\iint y \,\Delta U(x, y, z) \,dy \,dz}{\iint \Delta U(x, y, z) \,dy \,dz} \quad \text{and}$$
 (13a)

$$y_{\text{WC}} = \frac{\iint y \, \Delta U(x, y, z) \, dy \, dz}{\iint \Delta U(x, y, z) \, dy \, dz} \quad \text{and}$$

$$z_{\text{WC}}(x) = \frac{\iint z \, \Delta U(x, y, z) \, dy \, dz}{\iint \Delta U(x, y, z) \, dy \, dz}.$$
(13a)

Figure 4 shows the streamwise velocity of the full wake at five diameters downstream, along with the results from the wake tracking algorithm for all models. The variation in wake deficit shape shows that the wake center is a rather diffuse concept for asymmetric time-averaged wakes and cannot fully capture all effects at play in the complex flow field. For example, at that location, as will be shown in Fig. 5, while the RANS model underpredicts the wake deflection according to the wake tracking algorithm, it captures all the wake asymmetry due to yaw and veer. In contrast, at that specific location, the DWM model and PyWake give the closest estimate of the wake center as compared to LES, but the wake shape looks completely different. As we shall see later, for the inflow conditions presented here, that works mostly for the power predictions since the rotor mean velocity in the wake is still well matched. However, for the load predictions, matching the actual shape of the wake deficit becomes more important.

Figure 4. Streamwise time-averaged velocity at 5D downstream for all models for (top) zero, and (bottom) minus thirty degrees yaw misalignment. The white circles outline the rotor of the upstream turbine.

Figure 5 shows the location of the wake center determined by the wake tracking algorithm for all models and yaw angles.

Figure 5. Wake center position in (left) horizontal and (right) vertical as obtained from the wake-tracking algorithm of the time-averaged flow field for all models and yaw angles ( $\gamma$ ) at veer of  $\phi = 6^{\circ}$ .

For the lateral deflection due to yaw misalignment, all models capture at least the right sign for the deflection, but otherwise, there is a large spread in the results. RANS underpredicts the lateral deflection for all simulated angles except for the positive deflection angle  $\gamma=10^\circ$ , where it mildly overpredicts the deflection as compared to LES in the far wake. The authors speculate that this might be due to shortcomings of the turbulence model in the near wake, which are more amplified in yawed versus non-yawed flow. Averaged over all deflection angles, the DWM model using the Hill Vortex deflection model and a spatial averaging filter for the particle advection gives the closest results to LES. Fuga, a linearized RANS model, predicts lateral deflection values similar to those of the RANS model, with a stronger underprediction of the deflection for negative yaw angles. Finally, the PyWake model using the JiménezJiménez model gives more mixed results with no clear trend. For the largest yaw angle, both for the DWM and the PyWake results, the shape of the lateral deflection line is more curved in the near than the far wake, as compared to LES. The difference in vertical deflection shows an even larger spread between models than the lateral deflection. LES and RANS show similar trends with an upward deflection in the near wake and an up- or downward deflection in the far wake, depending on how substantial the yaw misalignment is. The vertical deflection in the DWM model is only driven by the tilt, and hence it shows a steady upward deflection, either under- or overpredicting as compared to LES in the far wake, with decent agreement in the near wake. Fuga predicts a downward deflection even for the non-yawed case






for two reasons: (i) Fuga does not include the tilt deflection yet, and (ii) likely an overdiffusion of the wake in the vertical direction. A Gaussian filter is applied to the Fuga predictions to get some turbulence mixing, since the turbulence closure in Fuga only considers vertical shear effects, and this filter is potentially too strong, leading the wake to diffuse too quickly (Ott and Nielsen, 2014). PyWake, similar to the DWM model, also predicts a steady upward projection of the wake, which is not visible in the LES reference data. Finally, there is potentially some coupling between the lateral and vertical deflection of the wake since the wake-averaged veer increases with height. Only the models that model veer can capture this.

To finish the discussion of the mean results for the single wake case, Fig. 6 shows the normalized power error of a hypothetical waked turbine for different downstream positions compared to LES. There are two types of hypothetical waked turbines: (i) ghost turbines whose loading is evaluated by running a standalone HAWC2 simulation using the wake planes as input and including the HAWC2 induction model, and (ii) actuator disks whose loading is evaluated by calculating the rotor-averaged velocity in the wake plane and then passing it through a power curve. For the DWM, the ghost turbines should give mostly the same results as if the turbine was actually placed there because the same induction model is used. However, for the LES, the induction is obtained by coupling the flow and the structural solver, so the ghost turbines and the real turbines will not yield the same results, but the trends should still be the same. A more detailed comparison between a ghost and a real turbine for the LES simulations will be given in Sect. 3.3. The aggregated errors are also shown on the side and at the bottom of the plot. As a general trend for all models, the error decreases with downstream distance for the aligned case, higher yaw angles lead to larger error, and for the yawed cases, the power tends to be underpredicted in the far wake. For the aligned case, the RANS model shows small far-wake errors of  $\sim 10\%$ , whereas underpredicted wake deflection in the misaligned cases increases the far-wake relative error to  $\sim 25\%$ , yielding an overall error across all cases of about 20%. Of all the models, the DWM shows the smallest total error, just shy of 10 %, and the maximum error hovers around 15 %. Fuga shows errors up to 300% in the near wake (note the y-axis scaling, see the figure caption for more explanation); for the aligned case, the far wake prediction error is below 10 %, but for the non-aligned cases, the error in the far wake maxes out at about 30%. Finally, the results from PyWake are off by up to 150 % in the near wake, but from 4D onwards, the error decreases to a maximum of around 25 % for all cases, even without modeling the veer.

The instantaneous wake and turbine loading properties will be presented for the LES and the DWM predictions since these are the only two models that predict dynamic results and not just mean properties.

A wake-tracking algorithm was applied to instantaneous flow slices at various downstream locations to dynamically compare
the wake behavior of the turbine. Figure 7 presents the power spectral density (PSD) and time series of the wake centerline
displacement for the aligned case at five rotor diameters downstream.

For the Dynamic Wake Meandering (DWM) model, two results are shown: the output from the wake-tracking algorithm applied to the flow field and the actual positions of the wake-defining particles, representing the "true" wake center as defined by the model. The particle-based wake center exhibits less motion than the one inferred from the tracking algorithm. This highlights the ambiguity and sensitivity of defining a wake center, especially when working with instantaneous flow slices.

No scientific consensus on the root cause of wake meandering has been reached in the literature, according to Yang and Sotiropoulos (2019). Several plausible mechanisms have been proposed: (i) advection by large-scale inflow structures - the

Figure 6. Relative power error  $\epsilon = (P-P_{LES})/P_{LES}$  as compared to LES for the first turbines as well as for several ghost turbines placed at different locations in the wake. For the first turbine, the LES reference is a HAWC2 turbine, and then a HAWC2 ghost turbine is used for the waked turbines. Results for all yaw angles and models are presented. The aggregated error is given by  $\epsilon_{RMS} = \sum \sqrt{\epsilon^2}$ . A symlog scale is used for the y-axis, which means that up to  $|\epsilon| 






In the PSD plot, the cutoff frequency  $f_C$  identified by Larsen et al. (2008) and Larsen and Lio (2025) - above which vortices are considered too small to influence wake meandering - is indicated. Also shown is the Strouhal number range  $St = \frac{fD}{U_{hub}} \in [0.2, 0.4], \text{ where shear layer instabilities and wake interactions have been reported in the literature.}$ 

Compared to the LES reference, the DWM model predicts less movement of the wake centerline, especially in the vertical direction. From the plots, it is clear that the LES and DWM signals are correlated. The correlation is stronger for the lateral wake deflection, likely because at five diameters downstream, the mean wake center deflection of the DWM is closer to the one from LES in the lateral than in the vertical direction (see 5). Nevertheless, despite its simplicity, the DWM model captures the overall amplitude of the meandering and the ratio between lateral and vertical wake oscillations reasonably well for this setup.

Finally, Fig. 8 shows the relative error of the Damage Equivalent Loads (DELs) computed for the selected moments over the entire simulation period for the ghost turbines placed in the wake of the freestanding turbine. The DELs were computed for the blade root flapwise moments, the tower bottom foreaft moments, and the tower top yaw bending moments. Those load channels were selected based on comparison with the literature. The errors for the DELs for the blade root bending moment max out at around 20 % with larger errors for smaller yaw angles or the fully aligned case. For the tower bottom foreaft DELs, the error also increases with smaller yaw angles and is largest for the aligned case, with a maximum relative error of around 60 %. The predictions for yaw bending moment at the top of the tower are very accurate and increase with increasing yaw angles, but the errors remain below around 10 %.

While the accuracy for the DWM for the prediction of the blade root moments and the yaw bending moment is satisfactory, the tower bottom moment predictions are very far off and warrant further investigation. In literature, similar comparisons between LES and DWM simulation results for the tower bottom foreaft DEL show the same trend. Branlard et al. (2024) plot the ratio of the tower DELs between the second and first row turbine for two IEA 15MW turbines placed 7D apart in stable and neutral conditions. For the stable case, the ratio between the DELs for the LES is similar to the ones presented for the LES here. In the paper, the DWM underpredicts the ratio by about 20 %, whereas in the case presented here, the DWM underpredicts the ratio by about 40 %. Likewise, Hanssen-Bauer et al. (2023) and Hanssen-Bauer et al. (2025) analyze a row of four NREL 5 MW turbines spaced 7.5D apart, both below and above rated conditions. Below rated conditions with a turbulence intensity (TI) of around 4.6 %, all DWM models underpredict the tower bottom DELs of the waked turbines by roughly 50%. Above rated conditions with a TI of 5%, the underprediction is about 30 % for the second-row turbine and increases up to nearly 100 % for the third and fourth rows. For the below-rated case, additional turbulence intensities of 9 and 12 % were tested: at 9 % TI, the tower DELs were underpredicted by roughly 10 %, while at 12 % TI they were predicted with reasonable accuracy. Furthermore, comparison between different DWM implementations and measurements on a large 6MW offshore wind turbine showed underprediction of up to 7-35% for tower tilt moments and up to 23% underprediction for tower torsion, which are different but related load channels (Bernard et al., 2024b). In this paper, the DEL for the second row turbine at 7.5Dis underpredicted by about 50 %. While the prediction error for the tower bottom foreaft DEL is larger than the reference cases in literature, it is plausible that the increase in error is due to the increase in size of the turbine.

**Figure 7.** Time series and Power Spectral Density (PSD) of the time-series of the wake centers of the LES and the DWM predictions at 5D downstream for the aligned case. The top left side shows the lateral, and the top right side shows the vertical wake displacement. The time series for the (middle) lateral and the (vertical) deflection are plotted. For the DWM, the PSD of the waketracking algorithm applied to the output flow slices and the actual wake particle position are shown. A rolling filter with a width of 20 time steps is applied for the averaged results.

**Figure 8.** Relative prediction error of the (ghost) turbine DELs evaluated with the DWM inflow as compared to the ghost turbines run with the LES inflow. The results are shown for (left) flapwise blade loads at the root, (middle) fore-aft tower loads at the bottom of the tower, and (right) yaw bending loads at the top of the tower. The errors are also evaluated for all yaw angles from top to bottom; see the plot labeling. At x/D = 0, a real DWM turbine is compared to a real LES turbine, and at x/D > 0, DWM ghost turbines are compared to LES ghost turbines.





## 3.3 Setup II: Two aligned turbines

The results for the second setup with two turbines spaced 5D apart are shown in this section. This setup has been added to see how the ghost turbines compare with an actual turbine for the LES, and to see if the models can capture the secondary steering.

Figure 9 shows the mean flow fields for the aligned and the steered case with a yaw offset of  $\gamma=-30^{\circ}$ . Qualitatively, all models capture the increase in the wake deficit for the second turbine and the asymmetry of the wake of the second turbine due to the partial wake of the upstream turbine. However, the DWM, Fuga, and PyWake predictions do not capture any further deflection of the wake of the second turbine due to secondary steering, since they do not model the lateral velocity deficit that produces the steering. Initially, in the DWM model, the velocity deficit from the Ainslie model was projected according to the yaw and tilt misalignment of the rotor, which led to some secondary steering; however, this led to a strong underprediction of the streamwise velocity deficits. Projecting the velocity deficit only according to the local flow deflection angle was also trialed, but this leads to a minimal lateral velocity deficit, which does not sufficiently steer the wake of the downstream turbine.

Figure 10 shows the relative error in the predicted mean power for three cases: the first turbine, the waked second turbine at x = 5D, and a ghost turbine at x = 10D. In this setup, results are presented for both real and ghost turbines.

For the first-row ghost turbine with LES inflow planes, the errors are about 10 % for the aligned case and 3 % for the yawed case. This indicates that the induction model in HAWC2 does not fully capture the inflow complexity. For the second-row ghost turbine, the aligned case error decreases to 5 %, while the yawed case error increases to 20 %. This increase suggests that the induction model becomes less accurate for more complex inflows, which is expected.

When comparing the real second-row turbine instead of the ghost turbine, the differences are smaller for the aligned case. Hence, the conclusions from the previous section remain valid for the other models as well. In the yawed case, all models previously overpredicted the power compared to LES. However, the errors are smaller relative to the real turbine, meaning that the apparent overprediction against the ghost turbine was partly an artifact of the ghost representation.

For the third-row turbine, compared again against an LES ghost turbine, all models underpredict the power, likely partially due to not capturing the secondary steering. Yet, as seen for the second row, the lower-fidelity models may in fact be closer to the "true" LES behavior than suggested by the ghost turbine comparison alone.

Finally, the total RMS error (shown in the right and bottom panels of Fig. 10) is smaller than in the ghost-only case. This is mainly because the power underprediction for the second turbine is less severe when compared to the real turbine, and because the dataset now contains fewer yawed cases, where all models generally perform worse.

In the previous section, three DEL channels in the DWM were compared to LES, where the tower bottom flapwise bending moments deviated by up to 50 % at five diameters downstream. Figure 11 now shows the DEL prediction errors for the second setup with two turbines, again including both real and ghost turbine results from the LES.

In contrast to the power predictions, where ghost turbines tended to overpredict relative to the real turbine, the loads show the opposite trend: ghost turbines underpredict the DELs compared to the real turbine.

For the blade root flapwise DEL, the prediction errors remain in the same order of magnitude, whether the LES reference is the ghost or the real turbine. For the tower bottom moments, however, the DWM error increases to roughly 75 %, since the

Figure 9. Time-averaged streamwise velocity at hub height for all models for (left) the aligned and (right) the yawed case with  $\gamma = -30^{\circ}$ .

LES ghost turbine itself deviates by about 25 % from the LES real turbine for this channel. For the tower-top yaw DEL, the sign of the error in the DWM remains unchanged regardless of whether the ghost or the real turbine is used as reference.

Overall, because loads are more complex than power predictions, there is no simple one-to-one relationship between the DWM accuracy relative to ghost and real turbines in the LES.

Finally, Fig. 12 shows the load spectra of the second turbine for the aligned case. Qualitatively, the spectra look similar between LES and DWM, with the highest peak at 1P and 3P for blade/yaw and tower loads, respectively. However, the peak magnitude is underpredicted in the DWM as compared to LES. The underprediction is larger for higher frequencies, i.e. 2P,

Figure 10. Relative power error compared to LES for the first two turbines of the setup and a ghost turbine placed at 10D. Results for two yaw angles and all models are presented. The dotted bars are HAWC2 ghost turbines, and the bars with circles are obtained from rotor-averaged values plus the power curve.

3P, and 6P. Since the tower bottom loads have the highest energy at 3P, the underprediction of the signal energy at higher frequencies in the DWM is consistent with the underprediction of the DEL for said channel.

**Figure 11.** Relative prediction error of the DELs for both ghost turbines with LES inflow planes and the DWM as compared to real turbines in the LES simulation. The results are shown for (left) flapwise blade loads at the root, (middle) fore-aft tower loads at the bottom of the tower, and (right) yaw bending loads at the top of the tower. The errors are evaluated for two yaw angles, see the labeling of the plot.

Figure 12. Load spectra for the waked turbine at 5D downstream of the upstream turbine for different yaw misalignment angles. The DELs are for (left) flapwise blade loads at the root, (middle) fore-aft tower loads at the bottom of the tower, and (right) yaw bending loads at the top of the tower.

# 4 Discussion

The comparison between models of different fidelity revealed consistent trends but also systematic discrepancies that point to model limitations.








The RANS model qualitatively captured asymmetric wake effects due to veer and yawed inflow, lateral velocity deficits, and secondary steering, but the magnitude of these effects was underpredicted. This led to growing errors in power prediction with increasing yaw angles, and an underprediction of wake deflection due to veer, even for the aligned case.

The DWM gave the most accurate power predictions overall, mainly due to the good performance of the wake deflection model, but it did not capture wake asymmetry. Since the inflow conditions tested here only produced moderate veer and curling, the accuracy is expected to decrease under more strongly sheared or curled conditions. For turbine loading, blade and yaw bending moments were predicted with reasonable accuracy, but tower bottom loads showed large errors (up to about 75 % in DEL for the second setup). These discrepancies are likely linked to neglecting non-axisymmetric wake effects in the Ainslie model, the underprediction of wake meandering, and simplifying assumptions in the DWM turbulence treatment.

Using a uniform spatial averaging filter for meandering may contribute to the underprediction of the wake meandering unrelated to shortcomings in the model hypothesis. Comparison of the PSD of the wake center line with and without the filter enabled reveal that this filter tends to overdampen for certain frequencies. Hence, a more advanced filter could improve the performance. For example, in FastFarm, a non-uniform spatial averaging filter based on the Jinc function is implemented (FastFarm).

For the wake asymmetry, alternative modeling approaches have been proposed in the literature. Branlard et al. (2022) introduced the Curled Wake model in the DWM framework implemented in FastFarm, which can predict not only lateral velocity deficits but also reproduce some of the curling in the wake. However, for this specific case, a preliminary implementation of the curled wake model in-house reproduced the curling of the wake from about 5D onwards, whereas in the LES dataset, the curling is only visible up to about 5D for the single turbine case and a yaw misalignment of  $\gamma = -30^{\circ}$ . Further, the curled wake model does not capture the effect of veer on the wake shape, which is more important, since if there is no wake deflection, accurately predicting the wake shape becomes more important. More broadly, fast mid-fidelity wake models that capture the combined effects of veer, shear, and yaw on both power and loads remain an important development area. Potential candidates include fast 3D parabolic solvers (Mittal et al., 2017), simple wake deformation models (Abkar et al., 2018), and data-driven surrogates trained on high-fidelity data (Andersen and Murcia Leon, 2023; Schøler et al., 2024).

Additionally, even though the DWM implementation in Dynamiks uses the updated WAT model from Branlard et al. (2024), the TI in the wake is missing key features: (i) the effect of shear on the TI is not modeled hence the TI is correct at hub height but under- or overpredicted at the top and bottom the wake, respectively, (ii) accumulation of the TI in the farm is not correctly captured. Keck et al. (2015) has proposed improvements for both these problems by connecting the wake TI to the Reynolds stress in the wake and shear in the inflow, but they are not implemented in Dynamiks yet. The plan for the future is to improve based on his work. If a non-axisymmetric deficit model is implemented as suggested in the previous paragraph, then the first problem would also be much easier to solve.

Among the steady-state engineering models, the Supergaussian wake model in PyWake using the Jiménez Jiménez model for deflection gave similar accuracy to RANS in the far wake. Inclusion of veer-induced deflection similar to the DWM and case-specific retuning of the near-wake region could potentially improve the results further. Fuga performed similarly to PyWake in the far wake but showed errors up to 300 % in the near wake and overall slightly larger errors than PyWake for all cases,

https://doi.org/10.5194/wes-2025-200 Preprint. Discussion started: 20 October 2025

© Author(s) 2025. CC BY 4.0 License.

despite modeling more of the physics. Including tilt deflection and potentially modifying the Gaussian smoothing applied to 500 the wake could improve the results.

Finally, the DWM, Fuga, and PyWake do not model lateral velocity deficits and thus cannot capture secondary steering. This leads to flawed results for yaw setpoint optimization. A straightforward improvement could be to include a lateral velocity deficit term following Zong and Porté-Agel (2020).

#### 5 Conclusions

This study compared power and load predictions for wake steering on a solitary IEA 22 MW turbine and a two-turbine row with 5D spacing, using models of varying fidelity under neutral ABL conditions. Overall, all models reproduced the qualitative power and load variation trends with yaw angle and downstream position. Quantitatively, however, substantial differences in accuracy were observed. The DWM provided the most accurate power predictions but heavily underestimated tower bottom loads for various reasons. RANS captured asymmetric wake effects but underpredicted their magnitude, leading to errors in power prediction. Steady-state engineering models (PyWake and Fuga) produced reasonable predictions in the far wake for all yaw angles, but could benefit from small model-specific improvements. None of the engineering models reproduced secondary steering since the lateral velocity deficits were not modeled. Several suggestions for further improvements to the models have also been discussed. The improvements focus on capturing wake asymmetry due to non-uniform inflow, mean wake deflection, and lateral velocity deficits, as well as better capturing the main physics behind wake meandering.

The main takeaway is that while the presented models can capture broad trends for the neutral boundary layer case presented here, their quantitative accuracy is limited for the aforementioned reasons. For practical applications, this means that optimization strategies relying on the tested models should be applied cautiously, especially for yaw offsets larger than 10 degrees. Further, the results may become less reliable in cases with stronger veer.

Code and data availability. The dataset can be made available upon request. Dynamiks and Fuga are open-source. Fuga is in the process of being made open-source. EllipSys3D and HAWC2 are available with a license.

Author contributions. JS generated the results for the DWM, did all the data processing, and wrote the initial draft of the paper. EH performed the LES. MPvdL performed the RANS simulations. LA provided a script for the generation of the PyWake/Fuga. MP helped with dev work on Dynamiks, PyWake, and Fuga. PER acquired the funds for the hours. All authors contributed to the finalization of the paper.

Competing interests. The authors have declared no competing interests.

Acknowledgements. This research has been supported by the SUDOCO project, which is funded through the European Union's Horizon Europe Programme under grant agreement No. 101122256, and by Equinor. All simulations were performed on the DTU cluster Sophia (DTU Computing Center, 2021). Artifical Intelligence was used during the preparation of this manuscript solely for improvements in readability, the rest of the conceptualization and work was performed by the authors themselves.

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
