# Peer review of "A multi-fidelity model benchmark for wake steering of a large turbine in a neutral ABL"

_Wind Energy Science, 2025_

## Referee Comment (RC1)

**Overall comments:**

This article compares modeling fidelity of several levels of reduced-order models versus LES results for both fully aligned and yaw-deflected wakes. The thoroughness of the examination of discrepancies between models is commended, and the manuscript represents a valuable contribution to the literature. Several improvements are suggested to increase the paper's quality as noted below.

**Detailed comments:**

Line 311 – for the reader's convenience, please indicate what was updated about the WAT model since there seems to have been an important change between the formulation used in the PyWake results and that used in the DWM results.

Figure 6 – this is an innovative way to show a lot of information at once. Where are the square and plus symbols that are defined in the legend used in this plot?

Figure 6 – it would obviously require some number of new LES simulations, but this figure might be *most* impactful if the reference was the actual two-turbine array in LES rather than the ghost turbine setup. If such a comparison or similar is already included further below in the next section, is the reason for keeping Figure 6 to highlight certain (in)accuracies of the ghost turbine method?

Line 288 – the discussion around Figure 7 is helpful, but a little bit more might be desired. The authors note that correlation is stronger for the lateral direction than the vertical one and with good reasoning, but two points of interest stand out in the lateral spectra that are not noted:

- 1. Low frequency content is higher in LES
- 2. High frequency content that nicely aligns with the f\_{St} region is higher in LES

Line 396 – please note relevant literature or remove this line

Lines 401 – considering the comparison of LES and DWM data in the spectral domain, the analyses in this paper seems positioned to some degree to address aspects of the reason for the underprediction of tower-base DELs by the DWM, but no comments are made. For instance, some authors have previously noted that tower-base fore–aft DELs are known to be highly sensitive to the accuracy of the wind spectrum.

Figure 9 – the authors have used a wake-tracking algorithm to track wake centers. Given that some differences in hub-height wake width are apparent between models, what about showing some more quantitative comparisons of wake width at hub height?

Line 477 – please provide a reference for the new tool introduced

**Technical comments:**

Table 1 – please remove the (a), (b), and (c) markings since this is a table not a figure

Equation 13a-b – what is the reason why 13b is a function of x while 13a is not?

Line 330 – "that works mostly...", what is that?

Line 391 – "see 5"?

Line 393 – can you mention that these errors are DWM errors relative to LES?

Figure 8 – I think these figures would be more readable if the two different legends had titles: something like "Comparison reference" and "Model to be comparison" or similar

Page 21 –what is the reason for the multiple sequential paragraphs of one or two sentences?